# Health-Related Quality of Life and Physical Activity in a Community Setting

**DOI:** 10.3390/ijerph18147301

**Published:** 2021-07-08

**Authors:** Marta Gil-Lacruz, Ana Isabel Gil-Lacruz, Paola Domingo-Torrecilla, Miguel Angel Cañete-Lairla

**Affiliations:** 1Health Science Faculty, University of Zaragoza, 50009 Zaragoza, Spain; 2School of Engineering and Architecture, University of Zaragoza, 50018 Zaragoza, Spain; anagil@unizar.es; 3Education Faculty, University of Zaragoza, 50009 Zaragoza, Spain; paoladt10@gmail.com (P.D.-T.); mcanete@unizar.es (M.A.C.-L.)

**Keywords:** health-related quality of life, physical exercise, education, neighbourhood

## Abstract

This article analyses how physical activity reinforces each of the dimensions (mental, physical, social, etc.) of the health-related quality of life concept. To that end, we determined whether this relationship is moderated by educational level and area of residence. The empirical part was based on data obtained from a cross-sectional survey carried out in the Casablanca neighbourhood (Zaragoza, Spain). The sample comprised 1083 participants aged between 25 and 84 years residing in the three residential areas of this neighbourhood: Viñedo Viejo, Las Nieves and Fuentes Claras. These three areas exhibit significant socio-economic differences in their population. The self-reported questionnaire included the following key information for this study: socio-economic characteristics (sex, age, educational level and area of residence) and health-related quality of life (WHOQOL-Brief: mental health, physical health, social relations and environment). The main results obtained from the descriptive statistics and regression systems were added. Playing a sport or undertaking some physical activity brings many health benefits, both physical and mental. The educational level and area of residence affect this relationship, such that the effects of physical activity are greater for those residents of Casablanca who have a higher educational level and/or live in more favoured areas of this neighbourhood. The results have also been discussed by sex and age group. Investing in innovative programmes in educational institutions and communities to acquire healthy habits and behaviour patterns that take into account socioeconomic differences in the population would be an advisable public health strategy.

## 1. Introduction

There is currently a large volume of scientific literature regarding the positive impact of physical activity on the health and quality of life of the population [1,2,3]. Indeed, the World Health Organisation has published a series of detailed guidelines and recommendations regarding its advisability in all age groups [4,5]. Despite this, fewer than a third of adults (≥18 years) comply with these recommendations [6,7]. Physical inactivity is slowly being recognised as a global pandemic [8,9].

Spain also mirrors these alarming trends [10]. Indeed, in 2017, only four out of every 10 people practised some form of sport. Around seven out of every 10 people reported walking daily, whereas only four out of every 10 undertook physical exercise during their daily activities (work, studies, housework, etc.) [11]. Moreover, this risk is not distributed equally across the different population-based groups. For example, studies such as those by Martínez-Gómez et al. [12] and Sánchez-Alcaraz and Gómez [13] reported that Spanish males performed twice as much physical activity as their female counterparts. Indeed, in 2015, only 14% of Spanish women complied with the recommended amount of daily physical activity, compared with 45% of males [14]. 

Inactivity during leisure time and a sedentary behaviour at work and home, motorised transport and the increasing use of technology are some of the important barriers to adopting a healthy lifestyle [15,16,17,18,19]. Other factors such as an unhealthy diet and increased intake of toxic substances have also contributed to physical inactivity becoming one of the leading causes of death by causing cardio- and cerebrovascular disease [20].

A population-based study of physical activity facilitates the design of measures aiming to enhance quality of life and improve health. Indeed, promoting sports activities amongst all age groups is much cheaper than having to cover the health costs associated with treating the diseases caused by physical inactivity [21].

This study has two main objectives: first, to determine how physical activity improves each dimension in the health-related quality of life concept (mental, physical, social, etc.) in a community-based sample; secondly, to study how sociodemographic variables such as sex, educational level and area of residence affect this relationship.

### Theoretical Background

Health-related quality of life (HRQOL) is defined as: “individuals’ perception of their position in life in the context of the culture and value systems in which they live and in relation to their goals, expectations, standards and concerns” [22]. HRQOL is a multidimensional concept that includes domains related to physical, mental, emotional and social functioning [23]. The development of these domains depends on a set of socioeconomic factors, with educational level playing a key role [24,25]. 

One of the possible definitions of physical activity within the context of society is based on the set of activities that people undertake—because they want to, without the influence of external pressures—in their spare time in order to amuse, entertain and improve themselves provided that no material benefit is gained [26]. As well as having a positive impact on the physical dimension of health-related quality of life, undertaking physical activity also affects other related variables such as academic performance, self-esteem or lower alcohol, tobacco and drug consumption [27,28]. The scientific literature shows that physical exercise enhances psychological well-being and may be a preventive and therapeutic factor in numerous medical conditions [29,30]. Physical activity is also associated, in a statistically significant manner, with mental health [31,32], active ageing, functional independence and personal autonomy [33]. Therefore, it may be concluded that there is a close relationship between health and exercise [34,35], and between exercise and health-related quality of life [31,36,37]. 

Further research on how these relationships are affected by various factors is needed. Controlling for sociodemographic characteristics is possible to design adjusted prevention and health-promotion programmes. In this regard, an analysis of sex-based differences in the adoption of healthy lifestyles is a priority, as the measurement of both well-being and physical activity is lower in women than in men [6,7,38,39,40]. 

Regarding gender differences, it is remarkable that, only for men, the greater intensity and duration of physical exercise undertaken is associated with a better perceived health [29,41,42]. Even in wealthy countries, gender differences in physical activity need to be carefully addressed, because they reflect socioeconomic inequalities [43,44,45]. 

As far as age is concerned, starting sports activities at an early age facilitates the consolidation of active lifestyles throughout a person’s life cycle [46,47,48,49]. This learning will follow different paths during this cycle depending on issues such as working status or educational level [50,51,52].

Schooling and educational level are key factors in the learning of active behaviours. Active behaviours and health status, in turn, affect educational level and employment opportunities [53,54]. Therefore, educational level promotes longevity and health-related quality of life, especially in the well-being dimensions related to mental health and perception of the environment [55,56,57]. 

The area of residence is also an important criterion for social differentiation that reflects social stratification [58,59,60,61,62,63,64]. In this regard, smaller residential areas tend to present higher rates of physical activity [65,66], especially amongst the elderly [67,68].

Given the importance of the context in structuring our lifestyles and living conditions, neighbourhoods tend to be the focal point for numerous programmes aiming to encourage physical activity by way of community schemes [6,7]. As such, the main strength of this article lies in the small spatial and community context in which the study was carried out and the database used to that end. The limited geographical location of this study allows the development of programmes that focus on the needs of the community, with the key motivation of enhancing citizens’ participation in health and education as factors with an important effect on quality of life. 

## 2. Materials and Methods

This study focuses on the environment by contextualising the relationship between physical exercise and the dimensions of HRQOL in a local community. The main hypothesis is that the higher the levels of physical exercise, the better the level of HRQOL. We grouped the results by education and area of residence to ensure that the estimated coefficients of physical exercise did not include the effects of these socio-economic characteristics. We also repeated estimations by subsamples based on gender and age cohorts in order to determine whether the effect of physical exercise on HRQOL total and dimensions level changes, in terms of either direction (positive/negative) or intensity, for different population groups.

### 2.1. Sample

The survey was carried out in the Casablanca neighbourhood (Zaragoza, Spain) due to its diversity, such that three socioeconomic classes coexist in a single neighbourhood (Fuentes Claras: upper-middle class, Viñedo Viejo: lower-middle/working-class area, and Las Nieves: upper-middle class). The design of the neighbourhood (roads, a railroad and a canal) delimit the different socio-economic spaces. 

Taking into account the Data from the Health Centre on the distribution of parameters by sex, age and area of residence, we designed the population clusters to be representative of the Casablanca neighbourhood. We recruited five key informants from our research team (Welfare and Social Capital, University of Zaragoza), five social agents (social workers from the Community Health Centre, Youth Centre and Social Centres) and ten students in training (Journalists, Social Workers, Sociologists and Psychologists). All these collaborators coordinated in trying to maintain the design of the original composition of the cluster. The main inclusion criteria were sex, age and residence in Casablanca. The survey was carried out in a random system in streets, houses and social centres.

The final sample consisted of 1083 individuals aged between 25 and 84 years old. This sample is representative of the universe, taking into account the normal distribution of the variables, a confidence level of 95.5, δ error of 3% and the sex distribution of the population aged between 25 and 84 years living in Casablanca. The participants gave their consent to take part in all the activities.

### 2.2. Instrument

The instrument allows us to control for a wide range of variables:Socioeconomic characteristics: sex, age, educational level and area of residence.Physical activity frequency: Likert scale from 1 (every day) to 7 (never practice).Health-related quality of life: measured using the WHOQOL-BREF [69]. This instrument was selected due to its international nature and good psychometric properties of reliability and performance in preliminary validity tests [22,50,56,70].

HRQOL takes into account four dimensions of health: physical (7 items), mental (6 items), social (3 items) and environment (8 items). All items are measured by using a Likert scale that goes from 1 (very negative) to 5 (very positive). Health-related quality of life variables are measured as means of their corresponding items. In a previous study, the Cronbach’s alpha reliability coefficients were: 0.85, 0.77, 0.61 and 0.80 for the physical, psychological, social relations and environmental domains, respectively [56].

Whereas we have local data on sex distribution, the information about residence zone has been estimated based on previous studies [71].

The sample comprised 1083 individuals aged between 25 and 84 years and was equally distributed by sex: 506 men (47%) and 577 women (53%) (See Table 1). Three age groups were taken into account: 33% of interviewees were aged between 25 and 44 years, 47% between 45 and 64 years and 20% between 65 and 84 years. The age and sex distribution showed that the ratio of young people was higher for men than for women (37% of young men versus 30% of young women), whereas the ratio of seniors was higher for women than for men (23% for female seniors versus 16% for male seniors). 

In general, people living in Casablanca did not report good levels of health (around 1.5 points for each dimension out of 5, where 5 is excellent) (See Table 2). The worst-valued dimension was the social one, followed by environmental, mental and physical. In general, women reported worse levels of health than men, except in the case of the social dimension. Self-perception of health also worsened with age, with the physical dimension exhibiting the largest age gap and the environmental dimension the smallest.

With regard to educational level, 37% of the sample had attended university and 33% had completed secondary education (44% of interviewers had university studies). The data show that educational levels improved considerably among age groups. In addition, while male seniors were more likely to have attended university (14%) than female seniors (7%), the situation was opposite for younger people: 58% of young women had tertiary studies compared with 55% of young men. Only 14% of interviewees exercised daily, 42% exercised weekly and 44% did not participate regularly in sports activities. Our findings show that physical exercise intensifies among age groups, especially among men. Thus, only 11% and 5% of young men and women, respectively, exercised daily, in contrast to 39% and 19% of senior men and women, respectively. People aged from 45 to 64 years were the most sedentary, with around 50% of this age group not exercising regularly. Finally, 65% of interviewees lived in Viñedo Viejo, 23% in Las Nieves and 12% in Fuentes Claras. With regard to sex differences, a higher percentage of female seniors lived in Viñedo Viejo (74%) compared to male seniors (60%). It is also remarkable that there was a higher rate of male seniors living in Fuentes Claras (13%) than female seniors (4%). (See Table 3).

### 2.3. Empirical Strategy

The empirical strategy was based on the analysis of the descriptive results of the dependent and explanatory variables. The WHOQOL-BREF dimensions (physical, mental, social and environmental) and their combination as one variable were selected as dependent variables. Physical exercise, educational level and area of residence were considered explanatory variables. Three levels of *p* (<0.01, <0.05, <0.1) were selected to explain the significance of these variables. Repeated estimations were carried out by sex and age in order to identify differences in effects among population groups.

To model gender and age differences in relation to HRQoL, we carried out estimations through a linear system of ordinary least squares (STATA command: regression, mfx). The main reason to select this empirical methodology is the quantitative nature of the dependent variables.

## 3. Results

Table 4 shows regression coefficients for physical exercise, educational level and area of residence for the general state of health of men and women living in Casablanca. The estimations only confirm that men aged between 45 and 64 years who exercise daily have better health than men of the same age range who do not exercise regularly. Daily and weekly exercise has a positive effect on the health of women also aged between 45 and 64 years, with daily exercise being more positive than weekly exercise. In general, the results provided in Table 4 exhibit poor statistical significance.

Table 5 shows the regression coefficients for physical exercise, educational level and area of residence for the state of health of men living in Casablanca. The main difference with Table 4 is that results are reported for all four health dimensions. The results obtained using this strategy are rich in terms of content and robust in statistical significance. In general, people who exercise report better levels of health for each dimension; in addition, it is important to exercise regularly and to do so daily. Although all men benefit from physical exercise, men aged between 45 and 64 years seem to benefit most. For men aged between 45 to 64 years, physical exercise exerts the strongest effect on social health, followed by physical, mental and environmental. For people aged between 25 and 44 years, empirical evidence on the positive effects of physical exercise on health is essentially limited to weekly exercise. Thus, while weekly exercise is positively correlated with physical, mental and environmental health, daily exercise only has positive effects on mental health. The situation is opposite, however, for people aged between 65 and 84 years. Thus, for this age group, there is strong empirical evidence of the positive effects of daily exercise on physical, mental and environmental health, while weekly exercise only has positive effects on physical health. The results were also analysed by educational level and area of residence. With regard to education, this factor exhibits the strongest effects for physical and mental health in men aged between 45 and 64 years, and for the social and environmental health of men aged between 25 and 44 years. There is little evidence for seniors, although male seniors with secondary education exhibited better physical, mental and environmental health than those with only primary education. Living in affluent areas seems to reinforce all four health dimensions for men aged between 45 and 64 years. In contrast, the evidence for young people in such areas is mixed, with this age group reporting worse mental and social health than young people living in modest areas. There is no robust empirical evidence for seniors.

Table 6 provides the same results as reported in Table 5 but for women living in Casablanca. There is only empirical evidence for the positive health benefits of exercising for women aged between 45 and 64 years. For this age group, it is not only important to exercise regularly but to do so daily. The strongest effect of daily exercise was observed for social health, followed by environmental, mental and physical health. With regard to education, this factor has the strongest effects for physical and mental health in women aged between 45 and 64 years, but also displays an important effect for their social and environmental health. Education is positively correlated with environmental health for women aged between 25 and 44 years, and also with their physical and mental health, although to a lesser extent. There is essentially no robust evidence for seniors. The estimated coefficients for residential area are not statistically significant, except in the case of young women, for whom living in an affluent area has a positive effect on social health, and female seniors, for whom living in an affluent area has a positive effect on environmental health.

## 4. Discussion

Our results based on a neighbourhood survey are in line with a previous literature review based, in most cases, on national representative surveys. Both analysis levels require further research on how the relationship between the health-related quality of life and physical activity is affected by sociodemographic factors. 

Men and women report similar trends in physical activities to those observed in other studies [6,7,38,39,40], women in Casablanca being more sedentary than men. These gender differences are more pronounced among seniors, because old ladies are characterized by low levels of education and income. Consequently, gender gaps explain that in Casablanca, senior women perceive their quality of life related to health worse than men, as it happens in other empirical studies [56,70,71,72]. 

Given that women have improved their educational levels and income status in recent decades, this research also confirms that these gender differences smooth for young cohorts. Therefore, following the argumentation line of previous studies, good perceived states of health reinforce physical exercise intensity and frequency among men, but also among women of young cohorts [29,41,42].

Under this background, it is essential to address socio-economic inequalities [44], because they determine differences in time allocation. For example, labour inactivity is associated to a large extent with sedentary roles among women [45]. Other studies carried out with retirees have revealed that physical activity decreases as they stop having a regular job [50] or, in contrast, increases as they have more free time available for leisure activities [51,52]. 

For seniors in Casablanca, the probability of making a commitment to physical activities as a leisure alternative also increases with education [50]. It is commonly accepted by the scientific community that undertaking physical activity implies knowledge, behaviour, skills and attitudes that may affect health and well-being [25]. Consistent with these theoretical hypotheses, our empirical results point out educational level as one of the predictors for longevity and health-related quality of life, especially in the dimensions related to mental and environmental health [55,56,57]. Educational level is an important force driving changes to, and the promotion of, physical activity [73,74].

Lastly, we also corroborate that the area of residence provides an important criterion for social stratification [58,59,60,61]. Quality of life in residential areas depends on several factors, such as neighbourhood relationships, the possibility for social support, the feeling of belonging, living conditions, work, transport, security and physical environment [62,63,64]. In this regard, smaller residential areas tend to present higher rates of physical activity [65,66], especially amongst the elderly [67,68].

Summarizing, our findings suggest that, in addition to being the subject of a behavioural study, physical activity and inactivity are related to the models of equality and inequality that currently prevail in our society [44]. 

Concerning strengths and limitations, we would like to highlight the two main strengths of this research. The first is that by combining the effect of sociodemographic variables such as sex, age, educational level and residence, it gives us a broader perspective of inequalities in health and healthy lifestyles, such as the practice of physical exercise. The diversity of the population and the social determinants of health are important struggles for health promotion, both at the research and intervention levels. Second, the fact that the sample is obtained in a community setting facilitates the design of preventive and promotional strategies for physical exercise based on the natural context.

Regarding limitations, and by extension, future lines of research, it would be of interest to perform a more in-depth study of the sex-based differences in physical activity by including other sociodemographic variables that may be affecting this relationship [45]. An alternative means of measuring physical activity not based on self-reporting [75] would also be advisable. The fact that our methodological design is cross-sectional makes it difficult to establish conclusions regarding the causal relationships between variables. A long-term follow-up would make extrapolation easier while also allowing the efficacy of the health-promotion initiatives to be analysed [76].

In any case, the benefits of physical activity represent an important promoting effect on health-related quality of life, perhaps even more so during the current COVID-19 pandemic [32].

## 5. Conclusions

In relation to the first objective of this research analysing physical activity in the sample and its impact on the health-related quality of life concept, our findings show that nearly half of the population interviewed do not undertake any kind of physical activity regularly, and that educational levels are high. Physical exercise intensifies with age, and younger cohorts are more qualified. Our results confirm that the effects of physical exercise on health are more accurate when estimating the four health dimensions (PhysicalHealth, MentalHealth, Relations and Environment) than when considering an aggregated health measure.

Secondly, when we study how sociodemographic variables such as sex, educational level and area of residence affect this relationship, we find that, in general, exercising is positive for men and women, although there are important differences by sex and age group. Residential area and, especially, education are two variables that should be controlled when running estimations.

## Figures and Tables

**Table 1 ijerph-18-07301-t001:** Population and sample composition by gender, age group and area of residence.

Area of Residence	Men (*n* = 506)			Women (*n* = 577)			Total
	25–44 Years Old	45–64 Years Old	65–84 Years Old	25–44 Years Old	45–64 Years Old	65–84 Years Old	
ViñedoViejo	122	152	50	100	176	101	701
FuentesClaras	18	35	11	23	40	6	133
LasNieves	45	51	22	51	56	24	249
Total	185	238	83	174	272	131	1083

**Table 2 ijerph-18-07301-t002:** Average median of the dependent variables and mean of explanatory variables (*n* = 1083).

Variables	Description	Total
		Mean
Dependent variables		
PhysicalHealth	Range values: 1 = no physical health—5 = excellent physical health	1.48
MentalHealth	Range values: 1 = no mental health—5 = excellent mental health	1.46
Relations	Range values: 1 = no relational health—5 = excellent relational health	1.43
Environment	Range values: 1 = no environmental health—5 = excellent environmental health	1.45
Independent variables		
Women	Dummy variable that informs if the individual is a woman (1) or a man (0).	0.53
Age_25–44	Dummy variable: 1 if the individual is aged between 25 and 44 years old, 0 otherwise.	0.33
Age_45–64	Dummy variable: 1 if the individual is aged between 45 and 64 years old, 0 otherwise.	0.47
Age_65–84	Dummy variable: 1 if the individual is aged between 65 and 84 years old, 0 otherwise.	0.20
Sport_Infrequent	Dummy variable: 1 if the individual works out occasionally or never, 0 otherwise.	0.44
Sport_Weekly	Dummy variable: 1 if the individual never works out weekly, 0 otherwise.	0.42
Sport_Daily	Dummy variable: 1 if the individual never works out daily, 0 otherwise.	0.14
Primary	Dummy variable: 1 if the individual has primary studies, 0 otherwise.	0.30
Secondary	Dummy variable: 1 if the individual has secondary studies, 0 otherwise.	0.33
Tertiary	Dummy variable: 1 if the individual has tertiary studies, 0 otherwise.	0.37
ViñedoViejo	Dummy variable: 1 if the individual lives in Viñedo Viejo, 0 otherwise.	0.65
FuentesClaras	Dummy variable: 1 if the individual lives in Las Nieves, 0 otherwise.	0.12
LasNieves	Dummy variable: 1 if the individual lives in Fuentes Claras, 0 otherwise.	0.23

**Table 3 ijerph-18-07301-t003:** Mean of the dependent and explanatory variables by sex and age (*n* = 1083).

Variables	Men (*n* = 506)			Women (*n* = 577)		
	25–44 Years Old	45–64 Years Old	65–84 Years Old	25–44 Years Old	45–64 Years Old	65–84 Years Old
Dependent variables						
PhysicalHealth	1.58	1.55	1.39	1.52	1.45	1.30
MentalHealth	1.52	1.49	1.43	1.49	1.42	1.35
Relations	1.45	1.40	1.33	1.53	1.43	1.35
Environment	1.46	1.49	1.45	1.45	1.43	1.43
Independent variables						
Sport_Infrequent	0.34	0.46	0.28	0.45	0.52	0.43
Sport_Weekly	0.54	0.38	0.30	0.49	0.36	0.34
Sport_Daily	0.11	0.14	0.39	0.05	0.10	0.19
Primary	0.05	0.17	0.66	0.05	0.38	0.82
Secondary	0.39	0.46	0.19	0.37	0.30	0.11
Tertiary	0.55	0.36	0.14	0.58	0.32	0.07
ViñedoViejo	0.66	0.64	0.60	0.57	0.64	0.74
FuentesClaras	0.10	0.15	0.13	0.13	0.15	0.04
LasNieves	0.24	0.21	0.27	0.29	0.21	0.18

**Table 4 ijerph-18-07301-t004:** Regression coefficients of health socio-economic determinants for men and women living in Casablanca (dependent variable: Total).

Variables	Men (*n* = 506)			Women (*n* = 577)		
	25–44 Years Old	45–64 Years Old	65–84 Years Old	25–44 Years Old	45–64 Years Old	65–84 Years Old
Sport_Infrequent	-	-	-	-	-	-
Sport_Weekly	0.122	0.078	0.059	−0.060	0.495 ***	−0.209
Sport_Daily	0.263	0.224 *	−0.188	0.304	0.659 ***	−0.082
Primary	-	-	-	-	-	-
Secondary	0.152	−0.135	0.171	−0.220	0.863 ***	−0.258
Tertiary	0.022	0.067	−0.423	−0.002	0.834 ***	−0.494
ViñedoViejo	-	-	-	-	-	-
FuentesClaras	0.206	0.177	0.615	−0.205	0.280	0.244
LasNieves	−0.061	0.075	−0.443	0.005	0.227	−0.427 **
Intercept	25.570 ***	25.651 ***	25.577 ***	25.791 ***	24.541 ***	25.572 ***

***, ** and * denote levels of statistical significance of 0.01, 0.05 and 0.10, respectively.

**Table 5 ijerph-18-07301-t005:** Regression coefficients of health socio-economic determinants for men living in Casablanca (*n* = 506).

Variables	PhysicalHealth			MentalHealth			Relations			Environment		
	25–44 Years Old	45–64 Years Old	65–84 Years Old	25–44 Years Old	45–64 Years Old	65–84 Years Old	25–44 Years Old	45–64 Years Old	65–84 Years Old	25–44 Years Old	45–64 Years Old	65–84 Years Old
Sport_Infrequent	-	-	-	-	-	-	-	-	-	-	-	-
Sport_Weekly	0.102 ***	1.149 ***	1.378 *	0.961 ***	0.930 ***	0.437	0.661	1.721 ***	0.217	0.764 ***	0.532 **	0.607
Sport_Daily	0.310	1.530 ***	1.497 **	0.869 *	1.534 ***	0.927 **	0.668	2.210 ***	0.769	0.037	1.216 ***	1.544 ***
Primary	-	-	-	-	-	-	-	-	-	-	-	-
Secondary	0.753	1.445 ***	1.373 *	1.119	0.828 **	1.645 ***	3.228 ***	0.456	0.390	1.599 **	1.104 ***	1.193 **
Tertiary	0.666	1.823 ***	−0.104	1.308 *	1.470 ***	0.948	3.327 ***	1.156 **	0.311	2.062 ***	1.728 ***	0.486
ViñedoViejo	-	-	-	-	-	-	-	-	-	-	-	-
FuentesClaras	0.639	0.694 *	0.153	0.496	0.780 **	−0.800	1.626 **	1.560 ***	−0.701	0.944 **	0.703 *	0.116
LasNieves	−0.447	0.003	−0.961	−0.827 **	0.653 **	−0.786	−1.541 ***	0.810 *	−0.654	−0.121	0.569 *	−0.635
Intercept	14.639 ***	13.389 ***	12.856 ***	13.594 ***	13.185 ***	13.707 **	11.156 ***	12.003 ***	13.108 ***	12.383 ***	13.127 ***	13.532 ***

***, ** and * denote levels of statistical significance of 0.01, 0.05 and 0.10, respectively.

**Table 6 ijerph-18-07301-t006:** Regression coefficients of health socio-economic determinants for women living in Casablanca (*n* = 577).

Variables	PhysicalHealth			MentalHealth			Relations			Environment		
	25–44 Years Old	45–64 Years Old	65–84 Years Old	25–44 Years Old	45–64 Years Old	65–84 Years Old	25–44 Years Old	45–64 Years Old	65–84 Years Old	25–44 Years Old	45–64 Years Old	65–84 Years Old
Sport_Infrequent	-	-	-	-	-	-	-	-	-	-	-	-
Sport_Weekly	−0.457	0.551 **	0.430	−0.003	0.517 **	0.182	−0.120	0.481	−0.452	0.325	0.691 ***	0.026
Sport_Daily	0.022	1.595 ***	0.144	−0.145	2.024 ***	0.695	−0.265	2.438 ***	0.270	−0.230	2.098 ***	0.745
Primary	-	-	-	-	-	-	-	-	-	-	-	-
Secondary	1.178	1.659 ***	1.388 *	1.199	1.295 ***	0.703	−0.280	0.938 **	0.283	2.136 ***	1.039 ***	−0.492
Tertiary	2.421 ***	2.412 ***	1.039	2.404 ***	1.777 ***	1.120	0.579	1.027 **	0.299	2.846 ***	1.773 ***	0.199
ViñedoViejo	-	-	-	-	-	-	-	-	-	-	-	-
FuentesClaras	0.904	0.423	−1.845	0.568	0.401	0.287	1.952 ***	−0.125	−0.942	0.705	1.173 ***	1.041
LasNieves	0.152	0.484	0.309	0.287	−0.010	0.235	−0.106	−0.167	−0.219	−0.005	0.367	−0.134
Intercept	13.376 ***	12.686 ***	12.596 ***	12.930 ***	12.852 ***	13.084 ***	14.911 ***	13.283 ***	13.643 ***	11.831 ***	12.768 ***	14.159 ***

***, ** and * denote levels of statistical significance of 0.01, 0.05 and 0.10, respectively.

## Data Availability

Not applicable.

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
