# Peer review of "Health-Related Quality of Life and Physical Activity in a Community Setting"

_ijerph, 2021, doi:10.3390/ijerph18147301_

Round 1
Reviewer 1 Report
My recommendations are the following:
Line 132, I recommend you to clarify what the direction parameter refers to.
Line 140 correct the age of the participants, it is between 25 - 84, not 5-84.
Section 2.1. I recommend you to mention how the selection of these subjects was made, which were the inclusion and exclusion criteria.
I recommend that the titles of tables 2,3 be rewritten, focused on the data presented, they are identical.
I recommend mentioning the α-Cronbach coefficient for questionnaires.
Mention the period when this study was conducted.
Mention that you repeat the estimates, under what conditions, when, how I recommend clarification.
I recommend you to mention the differences between the estimates.
Line 157 refers to several studies but mention only one bibliographic index, I recommend clarification.
I recommend that the Discussions section be mentioned separately from the conclusions.
I recommend that you mention the limitations and strengths of this study.
The mentions in the last section are very poor. Discussions should focus on correlating the data of the present study with previous studies.
Line 272-273 is not clear this conclusion, it is too vague. The conclusions are not focused on results.
In conclusion, the study has several shortcomings, the novelty consists only in the presentation of some results, some taking from other studies according to line 155-156. It does not have a new proposal to improve the targeted parameters.
Author Response
First at all, we would like to thanks the interesting review of referee 1. Without doubts his/her insightful recommendations will improve the quality of our research.
Line 132, I recommend you to clarify what the direction parameter refers to.
In order to specify better, the direction parameter new version is:
We also repeated estimations by sex and age group in order to determine whether the effect of physical exercise on HRQOL total and dimensions level changes, in terms of either direction (positive/negative) or intensity, for different population groups.
Line 140 correct the age of the participants, it is between 25 - 84, not 5-84.
We agree with the reviewer. It is a spelling mistake so we have replaced “5” for “25”.
Section 2.1. I recommend you to mention how the selection of these subjects was made, which were the inclusion and exclusion criteria.
We have added this information:
Taking into account the Data from the Health Centre on the distribution of parameters by sex, age and area of residence, we designed the population clusters to be representative of the Casablanca neighbourhood. We recruited five key informants from our research team (Welfare and Social Capital, University of Zaragoza), five social agents (social workers from the Community Health Centre, Youth Centre and Social Centres) and ten students in training (Journalists, Social Workers, Sociologists and Psychologists). All these collaborators coordinated trying to maintain the design of the original composition of the cluster. The main inclusion criteria were sex, age and residence in Casablanca. The survey was carried out in a random system in streets, houses and social centres.
I recommend that the titles of tables 2,3 be rewritten, focused on the data presented, they are identical.
New version of table 2 is: Mean of the dependent and explanatory variables (N = 1083)
New version of table 3 is: Mean of the dependent and explanatory variables by sex and age (N=1083).
I recommend mentioning the α-Cronbach coefficient for questionnaires.
Following your recommendation, we have included the following sentence in the main text:
In a previous study, the Cronbach´s alpha reliability coefficients were: 0.85, 0.77, 0.61, and 0.80 for the physical, psychological, social relations, and environmental domains respectively [56].
Mention that you repeat the estimates, under what conditions, when, how I recommend clarification.
With the expression “repeated estimates by sex and age” (line 131) we mean that we disaggregate the sample into different subsamples (a first subsample by sex: men and women; a second subsample by age cohorts: youth, adults and older adults). We apply the different statistical techniques considering these two variables to investigate their impact on HRQL and physical exercise.
In order to be more precise we change de expression ““repeated estimates by sex and age” by: “repeated estimates by subsamples based on gender and age cohorts”.
I recommend you to mention the differences between the estimates.
We are glad to confirm that tables 4 and 5 show the statistical differences between groups by sex and age. Explanation about these differences are integrated in the text.
Line 157 refers to several studies but mention only one bibliographic index, I recommend clarification.
We agree with the reviewer. In order to be more explicit in the use of references we have added three references and a sentence about the international use and psychometric properties of WHOQOL-BREF
Health-related quality of life: measured using the WHOQOL-BREF [69]. This instrument was selected due to its international nature and good psychometric properties of reliability and performance in preliminary validity tests [22, 50, 56, 70].
HRQOL takes into account four dimensions of health: physical (7 items), mental (6 items), social (3 items), and environment (8 items). All items are measured by using a Likert scale that goes from 1 (very negative) to 5 (very positive). In previous studies, the Cronbach´s alpha reliability coefficients were: 0.85, 0.77, 0.61, and 0.80 for the physical, psychological, social relations, and environmental domains respectively [56].
I recommend that the Discussions section be mentioned separately from the conclusions.
We have renamed the section 4 from “Discussion” to “Conclusions”.
I recommend that you mention the limitations and strengths of this study. The mentions in the last section are very poor. Discussions should focus on correlating the data of the present study with previous studies.
We agree with the reviewer that the "Strengths and Limitations" section is one of the most important parts of the document. Given the length of the article, we have included a new paragraph, as short as possible, to highlight the research strengths:
We would like to begin this section by highlighting the two main strengths of this research. The first is that by combining the effect of sociodemographic variables such as sex, age, educational level and residence, it gives us a more border perspective of inequalities in health and healthy lifestyles, such as the practice of physical exercise. The diversity of the population and the social determinants of health is an important struggle for health promotion, both at the research and intervention levels. Second, the fact that the sample is obtained in a community setting facilitates the design of preventive and promotional strategies for physical exercise based on the natural context.
Regarding limitations, and by extension future lines of research,…
Line 272-273 is not clear this conclusion, it is too vague. The conclusions are not focused on results.
Conclusions are based on results, so to make it clearer we have rewritten the paragraph as: Our results confirm that the effects of physical exercise on health are more accurate when estimating the four health dimensions (PhysicalHealth, MentalHealth, Relations and Environment) than when considering an aggregated health measure (Total).
In conclusion, the study has several shortcomings, the novelty consists only in the presentation of some results, some taking from other studies according to line 155-156. It does not have a new proposal to improve the targeted parameters.
We appreciate your comments

Reviewer 2 Report
To authors
This study analyzes the effects of physical activity on health-related quality of life from a variety of perspectives.
Considering the following points, I think that it will be a better research if you resubmit it.
Major revision
(1) LL.60-124.
I would like you to remove the parts that are not directly related to this research and make the description a little more concise.
(2) 4.Discussion and Conclusion:
Please describe in the order of the hypothesis of LL.54-58. For example, firstly,…..,secondly,……
Minor revision
(1) Table 2. Dependent variables:
Since the Likert result does not seem to be normally distributed, please describe the median value instead of the average value. Along with that, SD is unnecessary.
(2) Table 2. Independent variables:
Please express Dummy variables as real numbers, not average values. Along with that, SD is unnecessary.
(3) LL.285-292:
Please indicate that this part is “Limitation”.
Author Response
This study analyzes the effects of physical activity on health-related quality of life from a variety of perspectives. Considering the following points, I think that it will be a better research if you resubmit it.
The authors gratefully acknowledge the review of referee 2, which have helped us to improve the quality of the final paper.
Major revision
(1) LL.60-124. I would like you to remove the parts that are not directly related to this research and make the description a little more concise.
Due to the complexity of the topic, we are aware that many issues have been addressed in the main text. The main reason for doing this is to offer a broad scope of research in order to make the article more attractive and cover a heterogeneous audience. At least, we try to define and discuss the different variables included in the empirical framework, such as: Quality of life related to health and physical exercise and their sociodemographic determinants: sex, age, education, area of residence. However, we are open to deleting specific sentences if the reviewer still deems it necessary. In that case, we would appreciate if you could provide us with more details on which parts are redundant.
(2) 4.Discussion and Conclusion: Please describe in the order of the hypothesis of LL.54-58. For example, firstly,…..,secondly,……
We agree with the reviewer that this structure improve the reading, so we have rewritten the conclusions in order to follow this recommendation:
In relation to the first objective of this research analysing physical activity in the sample and its impact on health-related quality of life concept, our findings show that nearly half of the population interviewed do not undertake any kind of physical activity regularly, and that educational levels are high. Physical exercise intensifies with age, and younger cohorts are more qualified. Our results confirm that the effects of physical exercise on health are more accurate when estimating the four health dimensions (PhysicalHealth, MentalHealth, Relations and Environment) than when considering an aggregated health measure (Total).
As was the case in previous studies, the frequency with which physical activity is undertaken is important in this study, especially for the mental and physical dimensions of health-related quality of life [50,72].
Secondly, when we study how sociodemographic variables such as sex, educational level and area of residence affect this relationship, we find that in general, exercising is positive for men and women, although there are important differences by sex and age group. Residential area and, specially, education are two variables that should be controlled when running estimations.
Minor revision
(1) Table 2. Dependent variables: Since the Likert result does not seem to be normally distributed, please describe the median value instead of the average value. Along with that, SD is unnecessary.
We really understand the point of the reviewer. However, we think the problem does not lies on the statistic (average value) but a wrong definition of the dependent variables in Table 2. We have included range values to clarify the meaning of the variables, and in the main text we have included the following sentence: “Health related quality of life variables are measured as means of their corresponding items”. SD were deleted in table 2.
(2) Table 2. Independent variables. Please express Dummy variables as real numbers, not average values. Along with that, SD is unnecessary.
We are very sorry but we have not understood the meaning of real numbers. Do you mean for example to describe the mean of original variables? For example, the original variable of education has 3 possible values (1: Primary Education, 2: Secondary Education and 3: Tertiary Education). Do you want us to provide the mean of the original variable? The result 2.07 is less intuitive that the average values for dummy variables which allows us to easily observe that 30% of the interviewed have primary studies, 33% secondary studies, and consequently 37% have achieved tertiary studies. If this answer is not convincing enough, please let us know. SD were deleted.
(3) LL.285-292: Please indicate that this part is “Limitation”.
Taking into account this comment, we have included a new section “5. Strenghts and Limitations”, starting with the main two strengths of this research:
We would like to begin this section by highlighting the two main strengths of this research. The first is that by combining the effect of sociodemographic variables such as sex, age, educational level and residence, it gives us a more border perspective of inequalities in health and healthy lifestyles, such as the practice of physical exercise. The diversity of the population and the social determinants of health is an important struggle for health promotion, both at the research and intervention levels. Second, the fact that the sample is obtained in a community setting facilitates the design of preventive and promotional strategies for physical exercise based on the natural context.
Regarding limitations, and by extension future lines of research,...

Round 2
Reviewer 1 Report
My recommendation is to mention the discussion section as well. In this section refer to the results of your study in correlation with other studies.
At the end of the discussion section it is recommended to mention the limitations and strengths, not as a separate section. That is, these aspects will be section 4, and the conclusions section 5.
Author Response
Reviewer 1
We confirm that in this new version we have two separate sections for the discussion and main conclusions. In addition, we have merged the section devoted to strengths and limitation into the section discussion. Regarding the discussion, we have included the following paragraphs:
Our results based on a neighbourhood survey are in line of previous literature review based, in most cases, on national representative surveys. Both analysis levels require further research on how the relationship between the health-related quality of life and physical activity is affected by sociodemographic factors.
Men and women report similar trends in physical activities, as those observed in other studies [6,7,38-40], being women in Casablanca more sedentary than men. These gender differences are more pronounced among seniors, because old ladies are characterized by low levels of education and income. Consequently, gender gaps explain that in Casablanca, senior women perceive their quality of life related to health worse than men, as it happens in other empirical studies [56, 70-72].
Given that women have improved their educational levels and income status in last decades, this research also confirms that these gender differences smooth for young cohorts. Therefore, following the argumentation line of previous studies, good perceived states of health reinforce physical exercise intensity and frequency among men, but also among women of young cohorts [29,41-43].
Under this background it is essential to tackle socio-economic inequalities [44], because they determine differences on time allocation. For example, labour inactivity is associated to a huge extend to sedentary roles among women [45]. In fact, other studies carried out with retirees reveal that physical activity decreases as they stop having a regular job [50] or, in contrast, increases as they have more free time available for leisure activities [51,52].
For seniors in Casablanca, the probability of making a commitment to physical activities as a leisure alternative also increases with education [50]. It is commonly accepted by the scientific community that undertaking physical activity implies a knowledge, behaviour, skills and attitudes that may affect health and well-being [25]. Consistent with these theoretical hypothesis, our empirical results point out educational level as one of the predictors for longevity and health-related quality of life, especially in the dimensions related to mental and environmental health [55-57]. Educational level is an important force driving changes to, and the promotion of, physical activity [73,74].
Lastly, we also corroborate that the area of residence provides an important criterion for social stratification [58-61]. Quality of life in residential areas depends on several factors, such as neighbourhood relationships, the possibility for social support, the feeling of belonging, living conditions, work, transport, security, physical environment, etc. [62-64]. In this regard, smaller residential areas tend to present higher rates of physical activity [65,66], especially amongst the elderly [67,68].
Summarizing, our findings suggest that, in addition to being the subject of a behavioural study, physical activity and inactivity are related to the models of equality and inequality that currently prevail in our society [44].

Reviewer 2 Report
To authors
This study analyzes the relationship between H-R QOL and Physical Activity from the viewpoint of sex, age, educational level, and area of residence. Previous studies have shown that the relationship between H-R QOL and Physical Activity differs between gender (men and women), age(s), educational level(s), and area(s) of residence. Therefore, the text of LL.79-119 in this manuscript should be simplified to about 20 lines (about half) in total by making about 5 lines for each Socioeconomic characteristic (sex, age, educational level, and area of residence). Is it not?
I understand the other points.
Author Response
Reviewer 2.
Therefore, the text of LL.79-119 in this manuscript should be simplified to about 20 lines (about half) in total by making about 5 lines for each Socioeconomic characteristic (sex, age, educatonal level, and area of residence). Is it not?
Thank very much for clarifying us how to address this issue. We are glad to confirm that we have reduced the length of these paragraphs about half (from 41 lines to 21: LL.79-99).
